# A general mechanism of KCNE1 modulation of KCNQ1 channels involving non-canonical VSD-PD coupling

Xiaoan Wu [1], Marta E. Perez[1], Sergei Yu Noskov [2] & H. Peter Larsson [1✉]

Voltage-gated KCNQ1 channels contain four separate voltage-sensing domains (VSDs) and a pore domain (PD). KCNQ1 expressed alone opens when the VSDs are in an intermediate state. In cardiomyocytes, KCNQ1 co-expressed with KCNE1 opens mainly when the VSDs are in a fully activated state. KCNE1 also drastically slows the opening of KCNQ1 channels and shifts the voltage dependence of opening by >40 mV. We here show that mutations of conserved residues at the VSD–PD interface alter the VSD–PD coupling so that the mutant KCNQ1/KCNE1 channels open in the intermediate VSD state. Using recent structures of KCNQ1 and KCNE beta subunits in different states, we present a mechanism by which KCNE1 rotates the VSD relative to the PD and affects the VSD–PD coupling of KCNQ1 channels in a non-canonical way, forcing KCNQ1/KCNE1 channels to open in the fully-activated VSD state. This would explain many of the KCNE1-induced effects on KCNQ1 channels.

[1] Department of Physiology and Biophysics, Miller School of Medicine, University of Miami, Miami, FL, USA. [2] Centre for Molecular Simulation, Department of Biological Sciences, University of Calgary, Calgary, AB, Canada. ✉email: plarsson@med.miami.edu

The flow of K$^+$ ions through voltage-gated potassium (Kv) channels plays a crucial role in regulating cellular excitation in diverse cell types[1–3]. In the heart, the delayed rectifier potassium I$_{Ks}$ (KCNQ1/KCNE1) channel is one of the main contributors to the repolarization process of the ventricular action potential[4–6]. Dysfunction of KCNQ1/KCNE1 channels affects the cardiac action potential duration, which can cause Long QT syndrome, cardiac arrhythmias, and sudden cardiac deaths[6–8].

The KCNQ1/KCNE1 channel consists of both a pore-forming alpha-subunit called KCNQ1, also known as Kv7.1 or KvLQT1, and an auxiliary beta-subunit called KCNE1, also known as MinK (Fig. 1)[4,5,9]. KCNE1 modulates the KCNQ1 channel in several ways. For example, co-expression of KCNQ1 with KCNE1 slows the activation kinetics and shifts the voltage dependence of activation to more positive voltages in KCNQ1 channels[4,5]. KCNE1 association also induces a larger current amplitude and single-channel conductance, as well as the elimination of the inactivation in the KCNQ1 channel[10,11]. In sharp contrast, another auxiliary beta-subunit KCNE3 removes the voltage dependence of KCNQ1 current activation, locking the channel open in the physiological voltage range[12]. However, the mechanism underlying the modulation of the KCNQ1 channel by KCNE subunits is still not completely understood.

The KCNQ1 channel, like other Kv channels, is composed of four subunits (Fig. 1b)[11,13,14]. Each subunit contains six transmembrane segments, S1–S6 (Fig. 1). The S1–S4 segments in each of the four subunits together form a peripheral voltage-sensing domain (VSD) while the S5 and S6 segments from all four subunits together form a central pore domain (PD). KCNQ1 shows a domain-swapped configuration, which means that the VSD from one subunit is adjacent to the PD from the neighboring subunit (Fig. 1b). The S4 segment harbors several positively charged residues and serves as the main voltage sensor of the VSD[11,15,16]. Upon depolarization, S4 moves towards the extracellular side and opens the channel via a so-called VSD–PD coupling, also known as voltage-sensor-to-gate coupling or electro-mechanical coupling. A canonical mechanism proposed for Kv channels is generally based on that the interactions between the S4–S5 linker and the cytoplasmic end of the PD form the VSD–PD coupling and that the outward displacement of S4 segment pulls on the S4–S5 linker, which moves the lower end of S6 and opens the channel[17,18]. The Cui lab[19,20] found that each S4 in the KCNQ1 channel activates in two steps: from the resting state to an intermediate state, and then to a fully activated state. They showed that KCNQ1 channels mainly open in the intermediate S4 state. The authors also identified two different groups of interactions between the S4–S5 linker and PD contributing to the VSD–PD coupling when S4 is in the intermediate and activated states, respectively. KCNE1 was shown to change the VSD–PD coupling of KCNQ1 channels in that KCNQ1/KCNE1 channels can only open after S4s move to the fully activated state[19,20]. One possible explanation could be that KCNE1 suppresses the intermediate-open (IO) state of KCNQ1 such that KCNQ1/KCNE1 channels can only exhibit the activated-open (AO) state[20]. However, how KCNE1 alters the VSD–PD coupling of KCNQ1 channels is not fully clear. In addition, recent studies have reported a non-canonical VSD–PD coupling in a domain-swapped Kv channel (the Shaker K$^+$ channel) that is formed by interactions at the VSD–PD interface[21–23]. Whether this non-canonical VSD–PD coupling, in addition to the canonical interaction between S4–S5 linker and lower S6, occurs in KCNQ1 and/or KCNQ1/KCNE1 channels is not known.

In this study, we show that, at least a part of, the VSD–PD coupling is due to the interactions at the VSD–PD interface. Using voltage clamp fluorometry (VCF), we find that mutations (F232A and F279A) of conserved residues at the S4 and S5 interface changed the VSD–PD coupling by allowing KCNQ1/KCNE1 channels to open from the intermediate S4 state. Utilizing mutant cycle analysis (MCA), we find that functional interactions at the interface between S4 and S5 are formed in KCNQ1 channels only in the presence of KCNE1, suggesting that KCNE1 modifies the KCNQ1 channel by altering VSD–PD interactions.

Using a recent cryo-EM structure of human KCNQ1 and KCNQ1/KCNE3 channels[24], as well as KCNQ1 molecular models[25], we here propose a mechanism by which KCNE1 induces a rotation of the VSD relative to the PD in KCNQ1 channels, such that residues (F232 and F279) at the VSD–PD interface would clash if KCNQ1/KCNE1 channels would open when the VSD is in the intermediate state. This KCNE1-induced rotation would push VSD towards PD and limit the movement of PD, therefore forcing KCNQ1/KCNE1 channels to open only when the VSD is in the fully activated state which happens at more positive voltages. This mechanism could underlie the mechanistic basis of KCNE1 modulation of KCNQ1 channels, such as changes in the voltage dependence of opening and the ion selectivity. The non-canonical VSD–PD coupling at the VSD–PD interface in KCNQ1/KCNE channels proposed here could apply to other domain-swapped Kv channels in terms of the modulation by auxiliary subunits.

## Results

**F232A allows KCNQ1/KCNE1 to open before voltage sensors are in the fully activated state.** Using VCF, one study[26] showed that the F232A mutation at the top of the S4 segment (Fig. 1a) shifts the voltage dependence of the conductance-voltage (GV) relation to more negative voltages in KCNQ1/KCNE1 channels, but not the voltage dependence of the fluorescence-voltage (FV) relation. We hypothesized that this is because F232A changes the VSD–PD coupling of KCNQ1/KCNE1 channels by allowing the opening to the IO state. To test this hypothesis, we here examined the channel opening (by ionic current) and S4 movement (by fluorescence) of the F232A mutant channel using VCF. The mutant KCNQ1-C214A/G219C/C331A (from here on simply called KCNQ1) was used as the background for VCF experiments by us and other groups[19,26,27]. Endogenous cysteines at positions 214 and 331 were mutated to alanines to avoid the potential labeling by the fluorophore at these sites. Instead, a cysteine was introduced at position 219 in the extracellular S3–S4 linker of the

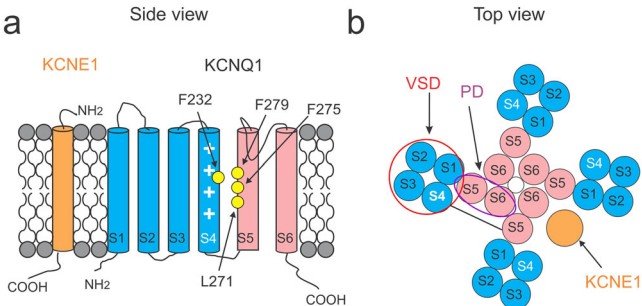

**Fig. 1 The topology of KCNQ1 and KCNE1. a** Schematic side view of one KCNQ1 subunit and one KCNE1 subunit in orange. S1–S4 form the voltage-sensing domain (VSD) in blue while S5–S6 the pore domain (PD) in pink. The white plus symbols in S4 represent the positive gating charges. Residues mutated to study the mechanism of VSD-PD coupling are indicated in yellow. **b** Schematic top view of a tetrameric KCNQ1 channel with one KCNE1 subunit. The number of KCNE1 varies from 1 to 4. One VSD (red circle) from one subunit is close to one PD (purple oval) from its neighboring subunit. S4 is linked to S5 by an S4-S5 linker within the same subunit.

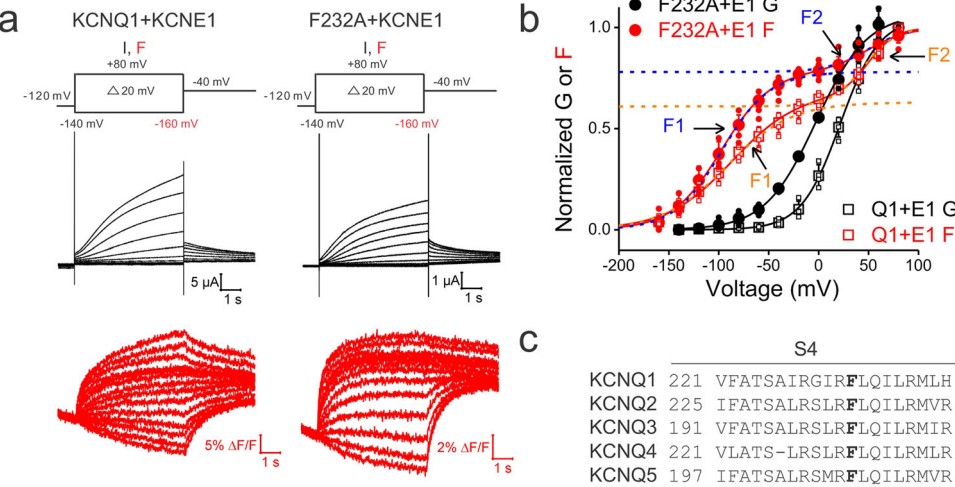

**Fig. 2 F232A allows KCNQ1/KCNE1 to open before voltage sensors are in the fully activated state. a** Current (black) and fluorescence (red) traces from oocytes expressing KCNQ1/KCNE1 and KCNQ1-F232A/KCNE1 channels. Cells are held at −120 mV and stepped to voltages between −140 mV (−160 mV for fluorescence) and +80 mV in +20 mV followed by a step to −40 mV. **b** Voltage dependence of currents (black) and fluorescence (red) from KCNQ1/KCNE1 (squares) and KCNQ1-F232A/KCNE1 (circles) channels. The GV relation was fitted with a single Boltzmann equation while the FV relation was fitted with a double Boltzmann equation. F1 and F2 indicate the first and second fluorescence components from KCNQ1/KCNE1 (orange dashed lines) and KCNQ1-F232A/KCNE1 (blue dashed lines) channels, respectively. **c** Sequence alignment of S4 in KCNQ family. F232 in KCNQ1 channels is conserved among other KCNQ channels. Data are shown as mean ± SEM. *N* is shown in Supplementary Table 1.

KCNQ1 channel for fluorophore labeling close to S4 to monitor S4 movement by the fluorescence changes.

Currents (in black) and fluorescence (in red) traces from KCNQ1/KCNE1 and KCNQ1-F232A/KCNE1 channels in response to a family of voltage steps are shown in Fig. 2a. In the wt KCNQ1/KCNE1 channel, the GV relation follows a single Boltzmann time course while the FV relation exhibits two components F1 and F2 (Fig. 2b)[19,26,28]. F1 represents the S4 movement from the resting state to the intermediate state and develops well before the channel opens. F2 represents the S4 movement from the intermediate state to the fully activated state and develops with channel opening (Fig. 2b)[28]. Thus, the wt KCNQ1/KCNE1 channel was suggested to open when S4 moves to the fully activated state[19,20]. Compared to wt, F232A shifted the GV relation and slightly shifted the F1 relationship to more negative voltages but not the F2 (Fig. 2b and Table. S1). The GV of F232A mutant channels occurred at more negative voltages and well before the development of the fully activated-S4 state (represented as F2) (Fig. 2b). This suggests that KCNQ1-F232A/KCNE1 can conduct even when S4 is in the intermediate state. Interestingly, our finding suggests that a single point mutation (F232A) at the VSD–PD interface changes the VSD–PD coupling by allowing gate opening from the intermediate S4 conformation.

**F232A changes the ion selectivity of KCNQ1/KCNE1.** To further test that F232A allows the KCNQ1/KCNE1 channel to open from the intermediate S4 state, the Rb+/K+ permeability ratio was measured. A previous report[29] found that the KCNQ1 channel shows a higher Rb+/K+ permeability ratio than the KCNQ1/KCNE1 channel. By using mutations that can lock the S4 in the intermediate or fully activated states, respectively, KCNQ1 channels were shown to have a higher Rb+/K+ ratio in the intermediate S4 state than in the fully activated S4 state[19]. Zaydman et al.[19] thus proposed that the higher Rb+/K+ permeability ratio corresponds to the IO state of KCNQ1 and the lower Rb+/K+ permeability ratio corresponds to the fully activated open state. Compared to KCNQ1 alone, KCNQ1/KCNE1 channel shows a lower Rb+/K+ permeability ratio, suggesting

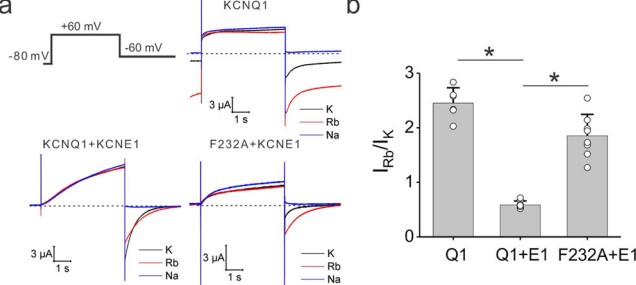

**Fig. 3 F232A changes the Rb+/K+ permeability ratio of KCNQ1/KCNE1 channels. a** Current traces from oocytes expressing KCNQ1, KCNQ1/KCNE1, and KCNQ1-F232A/KCNE1 channels under high external K+ (black), Rb+ (red), and Na+ (blue) concentration. Dashed lines indicate zero currents. Cells are held at −80 mV and stepped to +60 mV followed by a step to −60 mV. Note that KCNQ1 channels do not close completely at negative voltages. **b** Comparison of measured tail Rb+/K+ ratio from the KCNQ1 ($n = 6$), KCNQ1/KCNE1 ($n = 8$), and KCNQ1-F232A/KCNE1 ($n = 7$) channels. Data are shown as mean ± SEM, *$P < 0.05$. $P = 4.09349E-10$ between KCNQ1 and KCNQ1/KCNE1 channels. $P = 3.57806E-07$ between KCNQ1/KCNE1 and KCNQ1-F232A/KCNE1 channels.

that KCNQ1/KCNE1 only opens from the fully activated S4 state[19].

If KCNQ1-F232A/KCNE1 channels open with S4 in the intermediate state, one would expect these mutant channels to exhibit a higher Rb+/K+ permeability ratio than wt KCNQ1/KCNE1 channels. Figure 3 shows that unlike wt KCNQ1/KCNE1 that exhibited a Rb+/K+ ratio of 0.58 ± 0.07, the F232A mutant channel exhibited a significantly higher Rb+/K+ ratio of 1.85 ± 0.39 (Fig. 3b), although it was not as high as in KCNQ1 alone (2.45 ± 0.28) (Fig. 3b). Yet, the higher Rb+/K+ ratio from KCNQ1-F232A/KCNE1 channels indicates that F232A allows the KCNQ1/KCNE1 channel to open from the intermediate S4 state. Therefore, F232, conserved in the KCNQ family (Fig. 2c), might play an important role in the VSD–PD coupling of KCNQ1/KCNE1 channels by preventing the opening from the intermediate S4 state. The F232 residue, which localizes in the

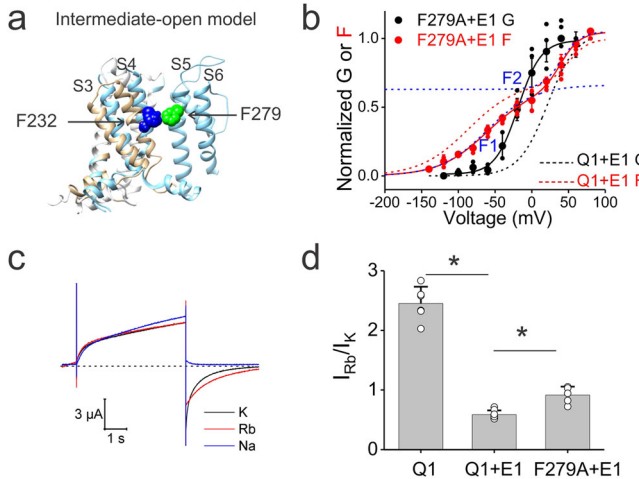

**Fig. 4 F279A also allows KCNQ1/KCNE1 to open in the intermediate S4 state. a** Overlay of an NMR VSD structure (gray) (PDB: 6MIE) and a cryo-EM structure (blue) (PDB: 6UZZ) of KCNQ1 channels. The NMR structure was suggested to be in the intermediate state and the cryo-EM structure was suggested to be in the fully activated state. F232 (blue) in S4 is close to F279 (green) in S5 in the intermediate VSD conformation. Only S3, S4, S5, and S6 are shown for clarity. **b** Voltage dependence of currents (black) and fluorescence (red) from KCNQ1/KCNE1 (dashed lines for comparison) and KCNQ1-F279A/KCNE1 (circles) channels. F1 and F2 indicate the first and second fluorescence components from KCNQ1-F279A/KCNE1 channels (blue dashed lines). *N* is shown in Supplementary Table 1. **c** Current traces from oocytes expressing KCNQ1-F279A/KCNE1 channels under high external $K^+$ (black), $Rb^+$ (red), and $Na^+$ (blue) concentration in response to voltage protocols indicated in Fig. 3. The dashed line indicates zero currents. **d** Comparison of measured tail $Rb^+/K^+$ ratio from the KCNQ1 ($n = 6$), KCNQ1/KCNE1 ($n = 8$) and KCNQ1-F279A/KCNE1 ($n = 5$) channels. Data are shown as mean ± SEM, *$P < 0.05$. $P = 4.09349E-10$ between KCNQ1 and KCNQ1/KCNE1 channels. $P = 0.00015$ between KCNQ1/KCNE1 and KCNQ1-F279A/KCNE1 channels.

middle of the S4 and seems important for the VSD–PD coupling of KCNQ1/KCNE1 channels, might represent a distinct VSD–PD coupling mechanism that differs from the canonical interactions between the S4–S5 linker and lower S6[17,18].

**F279A also allows KCNQ1/KCNE1 to open in the intermediate S4 state.** As F232A might change the VSD–PD coupling in the intermediate S4 state, we hypothesized that mutations of residues in contact with F232 in the intermediate state also change the VSD–PD coupling in the intermediate S4 state. Since there is no structure of the VSD and PD of the KCNQ1 channel in the IO state, we created our own KCNQ1 IO model by combining an NMR structure of the VSD (suggested to be in an intermediate state[30]) and the PD from a cryo-EM structure of the KCNQ1 channel (suggested to be in the fully activated open state[24]). PDB of this model is provided in the supplementary information (Supplementary Data 1). In our intermediate state model, residue F279 in S5 from the neighboring subunit seems closed to F232 and is likely to form interactions with it (Fig. 4a). A previous study[26] found that these two phenylalanines can form a disulfide bond upon depolarization when mutated to cysteines and that these two residues are functionally coupled with each other in KCNQ1/KCNE1 channels. In addition, F232A and F279A were both shown to have GVs that developed before the second FV component[26]. Since F232A mutant channels conduct when S4 reaches the intermediate state (Fig. 2b), we reasoned that F279A mutant channels can also open from the intermediate S4 state. To

test this idea, the VCF and the $Rb^+/K^+$ ratio experiments were conducted for the F279A mutation. F279A shifted the GV relation to more negative voltages and slightly shifted the FV relation (Fig. 4b), which is in line with the previous study[26]. Furthermore, the current activation of KCNQ1-F279A/KCNE1 started at more negative voltages than those for transitioning into the fully activated-S4 state (represented as F2) (Fig. 4b and Table. S1), as if the F279A mutant channel can conduct even when S4 is in the intermediate state. The F279A mutant displayed a $Rb^+/K^+$ ratio ($0.91 \pm 0.14$) significantly higher than wt KCNQ1/KCNE1 channels ($0.58 \pm 0.07$) (Fig. 4d), indicating that the mutant channel changes the ion selectivity and thus exhibits a fraction of open channels in the intermediate S4 state. Therefore, these results support our hypothesis that similar to F232A, F279A changes the VSD–PD coupling by opening from the intermediate S4 state in KCNQ1/KCNE1 channels. This suggests that F279 at the top of S5 plays an important role in the VSD–PD coupling of KCNQ1/KCNE1 channels by preventing the opening from the intermediate S4 state.

**F232A and F279A change the VSD–PD coupling in KCNQ1/KCNE1 but not in KCNQ1 alone.** As both F232A and F279A change the VSD–PD coupling of KCNQ1/KCNE1 channels by allowing the channel to open when S4 is in the intermediate state, the next question is whether the mutants could change the VSD–PD coupling of KCNQ1 channels expressed alone. Figure 5 shows the channel opening and S4 movement from wt and mutant (F232A and F279A) KCNQ1 channels without KCNE1 association. Similar to KCNQ1/KCNE1 channels, the wt KCNQ1 FV relation exhibited two components, F1 and F2. F1 indicates the S4 movement from the resting state to the intermediate state and F2 indicates the S4 movement from the intermediate state to the fully activated state[31,32]. F1 correlated with the GV relation suggesting that KCNQ1 opens when S4 is in the intermediate state (Fig. 5b), as shown previously[19,32]. Likewise, the FV relation of F232A (Fig. 5b) and F279A (Fig. 5c) showed two components, where the F1 overlapped with the GV relation (Table. S1). This suggests that without KCNE1 association, F232A and F279A do not change the VSD–PD coupling and open in the intermediate S4 state. We also found that both F232A and F279A do not change the $Rb^+/K^+$ ratio of KCNQ1 channels in the absence of KCNE1 (Fig. S1b), suggesting that these two mutations, similar to wt KCNQ1 channels[19], open in the intermediate state. Taken together, F232A and F279A change the VSD–PD coupling in KCNQ1 only in the presence of KCNE1.

**F232 and F279 are functionally coupled in KCNQ1/KCNE1 but not in KCNQ1.** So far, we have shown that F232A and F279A mutant channels have phenotypes (GV, FV relations and $Rb^+/K^+$ ratio) as if they allow channel opening from the intermediate S4 state. One possibility is that F232 and F279 functionally interact with each other in KCNQ1/KCNE1 channels when S4 is in the intermediate state. This interaction might be important for the intermediate VSD–PD coupling of KCNQ1/KCNE1 channels. To test this idea, we performed a thermodynamic MCA, which is a powerful approach to assess interactions between residues in ion channels[33]. The GV relation of the KCNQ1/KCNE1 channel was used to evaluate the change in the free-energy difference ($\Delta\Delta G$) between the closed state and the open state caused by the mutations. If the two residues do not functionally interact, the sum of the free-energy changes of the two single mutations will be equal to the free-energy change of its double mutation (Fig. 6a). That means the mutational effects of two single mutations are additive. Conversely, if the two residues interact, the sum of the free-energy changes of the two single mutations will be different from the free-energy

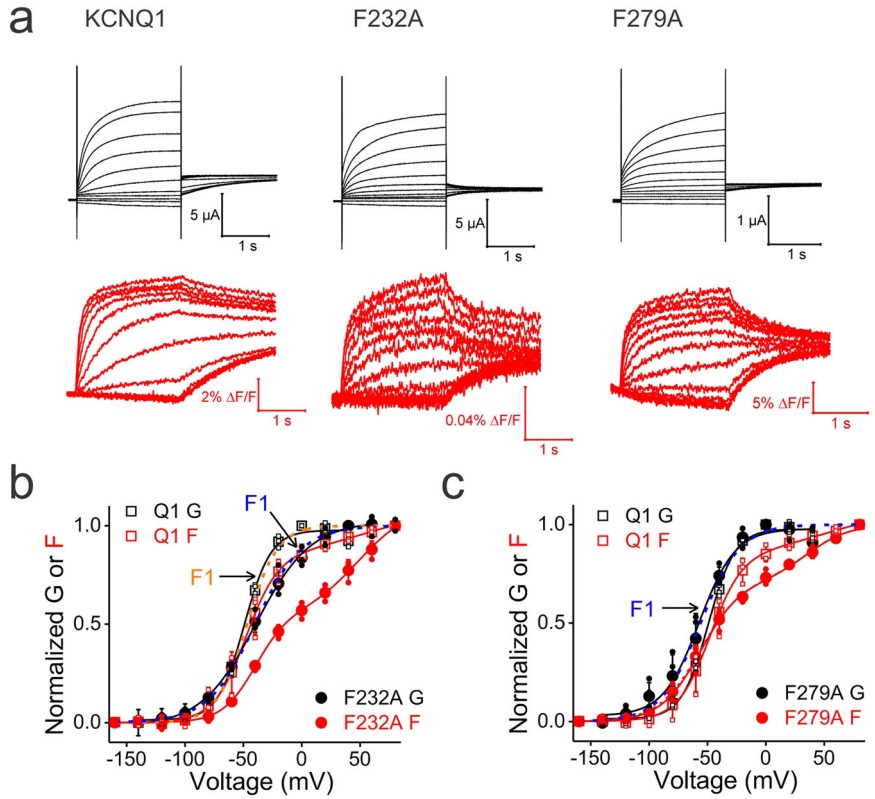

**Fig. 5 F232A and F279A do not change the VSD–PD coupling of KCNQ1 in the absence of KCNE1. a** Current (black) and fluorescence (red) traces from oocytes expressing KCNQ1, KCNQ1-F232A, and KCNQ1-F279A channels without KCNE1 association. The voltage protocol is indicated in Fig. 2. **b** Voltage dependence of currents (black) and fluorescence (red) from KCNQ1 (squares) and KCNQ1-F232A (circles) channels. F1 indicates the first fluorescence component from KCNQ1 (orange dashed line) and KCNQ1-F232A (blue dashed line) channels. Only F1 is shown for clarity. **c** Voltage dependence of currents (black) and fluorescence (red) from KCNQ1 (squares) and KCNQ1-F279A (circles) channels. F1 indicates the first fluorescence component from KCNQ1-F232A channels (blue dashed line). Data are shown as mean ± SEM. $N$ is shown in Supplementary Table 1.

change of the double mutation (Fig. 6b), which means the mutational effects are not additive. The difference in these free-energy changes is called the coupling energy. A coupling energy greater than 0.89 kcal/mol is considered to be significant[34,35].

Figure 6c shows the GV relations of F232A, F279A, and F232A/F279A of KCNQ1 in the presence of KCNE1. Compared to wt KCNQ1/KCNE1 channels, all three mutations shifted the GV in the negative direction. However, F232A/F279A showed a smaller GV shift compared to F279A. The free-energy difference of F232A/F279A ($\Delta\Delta G = 0.65$ kcal/mol) was smaller than either F232A ($\Delta\Delta G = 0.79$ kcal/mol) or F279A ($\Delta\Delta G = 1.39$ kcal/mol), suggesting that the mutational effects of F232A and F279A are not additive. The coupling energy calculated for F232A and F279A is ~1.54 kcal/mol, which suggests a strong interaction between F232 and F279 in KCNQ1/KCNE1 channels. Given that F232 and F279 are close in our IO model, the results suggest that F232 and F279 interact in the intermediate S4 state in KCNQ1 channels with KCNE1.

We also tested whether or not F232 and F279 are functionally coupled in KCNQ1 in the absence of KCNE1 (Fig. 6d). The free-energy difference of F232A ($\Delta\Delta G = 1.34$ kcal/mol) and F279A ($\Delta\Delta G = 0.41$ kcal/mol) approximates that of F232A/F279A ($\Delta\Delta G = 1.35$ kcal/mol), with a coupling energy of ~0.40 kcal/mol, which suggests that F232 and F279 do not strongly interact in KCNQ1 channels expressed alone. Therefore, the functional coupling between F232 in S4 and F279 in S5, as part of the VSD–PD interface, is formed in the KCNQ1 channel only in the presence of KCNE1. This suggests that F232 and F279 are sufficiently close to each other in the intermediate state only in the presence of

KCNE1, and that KCNE1 might directly or indirectly act on the F232–F279 interaction.

**KCNE1 is proposed to induce a rotation of the VSD relative to the PD and inhibit the IO state of KCNQ1.** Our data suggest that KCNE1 alters the interaction between F232 and F279 at the S4–S5 interface to change the VSD–PD coupling. However, in the human KCNQ1/KCNE3 cryo-EM structure (PDB: 6V01)[24], the distance between F232 and KCNE3 (a paralog of KCNE1) seems too far for direct interaction, and F232 seems not to even point towards KCNE3. Since previous studies have suggested that KCNE1 and KCNE3 share a similar location relative to KCNQ1[9,24], KCNE1 might not directly interact with F232 in S4.

How does KCNE1 then indirectly affects the F232-F279 interaction? The Mackinnon lab[24] has shown that KCNE3 induces a 7° counterclockwise rotation of the whole VSD relative to the PD of KCNQ1 channels, which allows KCNE3 to fit in between S1 and S6[24,36]. If KCNE1 were to rotate the VSD in a similar manner as KCNE3, then F232 would move relative to the PD and the F232–F279 interaction would be indirectly affected by KCNE1 (Fig. 7). We propose that in the absence of KCNE1, the larger space between S4 and S5 would allow wt KCNQ1 channels to open from the intermediate state (Figs. 7a, 7b, and S2e). However, upon KCNE1 association with KCNQ1 and the proposed rotation of the whole VSD, S4 in the VSD, and S5 in the PD from two different subunits would get closer together and the side chains of S4 and S5 would clash (Figs. 7c, d, and S2k). The compression of the interface between S4 and S5 would

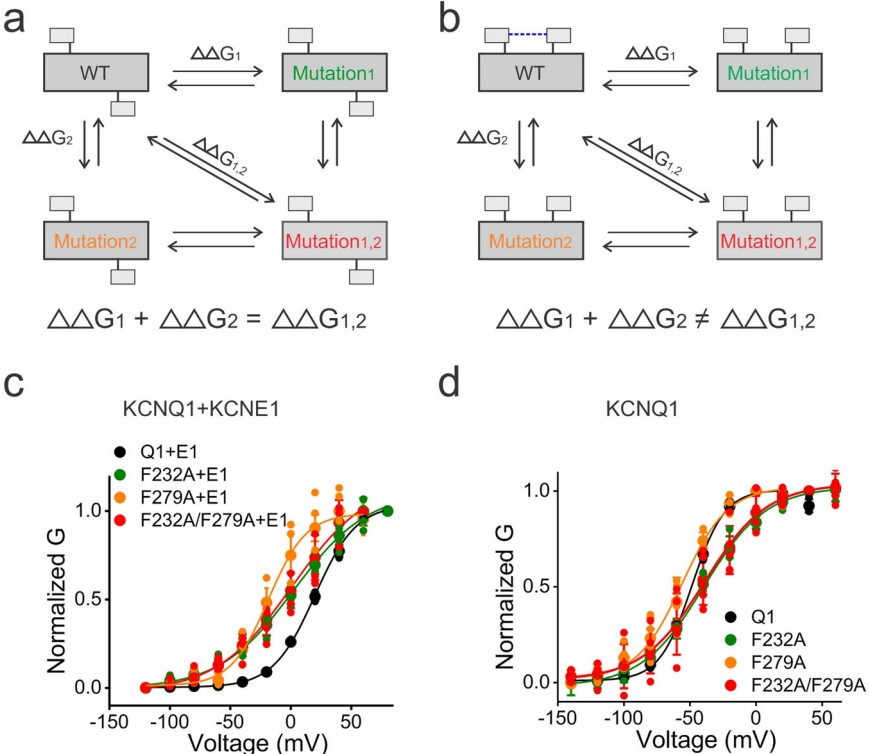

**Fig. 6 Mutant cycle analysis suggests that F232 and F279A interact in KCNQ1/KCNE1 channels but not in KCNQ1 channels alone. a** If mutant 1 (green) and mutant 2 (orange) are functionally independent of each other, the sum of the free-energy changes of the two single mutants will be equal to the free-energy change of double mutant 1,2 (red). **b** If mutant 1 (green) and mutant 2 (orange) are functionally interacting with each other (indicated as a blue dashed line), the sum of the free-energy changes of the two single mutants will be not equal to the free-energy change of double mutant 1,2 (red). **c** Voltage dependence of current activation from KCNQ1+KCNE1 (black, $n = 4$), KCNQ1-F232A/KCNE1 (green, $n = 4$), KCNQ1-F279A/KCNE1 (orange, $n = 3$), and KCNQ1-F232A-F279A/KCNE1 (red, $n = 5$) channels. **d** Voltage dependence of current activation from KCNQ1 (black, $n = 4$), KCNQ1-F232A (green, $n = 3$), KCNQ1-F279A (orange, $n = 4$), and KCNQ1-F232A-F279A (red, $n = 5$) channels without KCNE1 association. The voltage protocol is indicated in Fig. 2. Data are shown as mean ± SEM.

inhibit the movement of the PD to open the gate when S4 is in the intermediate state (Figs. 7c, d, and S2k). This would force KCNQ1/KCNE1 channels to open only when S4 is in the fully activated state at more positive potentials and suggest a mechanism by which KCNE1 greatly shifts the voltage dependence of current activation of KCNQ1 channels to more positive voltages (Fig. S2l). The clash between F232 and F279 is prevented when these residues are mutated to a smaller amino acid, such as alanine, so that the IO state of KCNQ1 is now allowed even in the presence of KCNE1. Our proposed model, in which KCNE1 compresses S4 towards S5 by a VSD rotation is consistent with evidence[26,37,38] showing that KCNE1 reduces the flexibility of the top part of the S4–PD interface. It also explains why the functional coupling between F232 in S4 and F279 in S5, as part of the VSD–PD interface, is formed in the KCNQ1 channel only in the presence of KCNE1.

**The KCNE1-induced rotation of the VSD affects the VSD–PD interaction in the resting S4 state.** We next tested whether VSD–PD interactions when S4 is in the resting state are also affected by a KCNE1-induced rotation of the VSD. In a recent resting-closed model of KCNQ1/KCNE1, F232, and L271 are physically close to each other when S4 is in the resting state (Fig. 8a)[25]. Using MCP, we found that F232 in S4 and L271 in the neighboring S5 interact strongly in the presence of KCNE1 but not in the absence of KCNE1. Figure 8b shows the effects of F232A, L271A, and F232A/L271A on the GV relation of KCNQ1 in the presence of KCNE1. The free-energy difference of F232A/

L271A ($\Delta\Delta G = 0.47$ kcal/mol) was smaller than either F232A ($\Delta\Delta G = 0.79$ kcal/mol) or L271A ($\Delta\Delta G = 0.77$ kcal/mol), suggesting that the mutational effects of F232A and F279A are not additive. The coupling energy calculated for F232A and L271A is ~1.08 kcal/mol, which suggests a strong interaction between F232 and L271 in KCNQ1/KCNE1 channels. However, the F232–L271 interaction is not strong in KCNQ1 channels expressed alone because the free-energy difference of F232A ($\Delta\Delta G = 1.34$ kcal/mol) and L271A ($\Delta\Delta G = 1.51$ kcal/mol) seem additive ($\Delta\Delta G = 2.00$ kcal/mol for F232A/L271A) and the coupling energy is ~0.85 kcal/mol (Fig. 8c). This indicates that the interaction between F232 and L271 is weaker in KCNQ1 without the KCNE1 association. However, if KCNE1 imposes a rotation of the VSD compared to KCNQ1 expressed alone as illustrated in Fig. 7, the interface between S4 and S5 would be compressed. F232 then would be closer to interact with L271 in the resting S4 state. The MCA between F232 and L271 in KCNQ1 with and without KCNE1 thus supports our model about how KCNE1 affects the VSD and then the PD through the VSD–PD interface.

**F275A does not allow KCNQ1/KCNE1 to open in the intermediate S4 state.** In our KCNQ1 IO model (Supplementary Data 1), F275 in S5 from the neighboring subunit seems physically close to F232 (Fig. S3a). We, therefore, tested whether a potential F232–F275 interaction contributes to the non-canonical VSD–PD coupling in KCNQ1/KCNE1 channels. F275A shifted the GV relation to more positive voltages and shifted the FV relation to more negative voltages (Fig. S3b). Furthermore, the

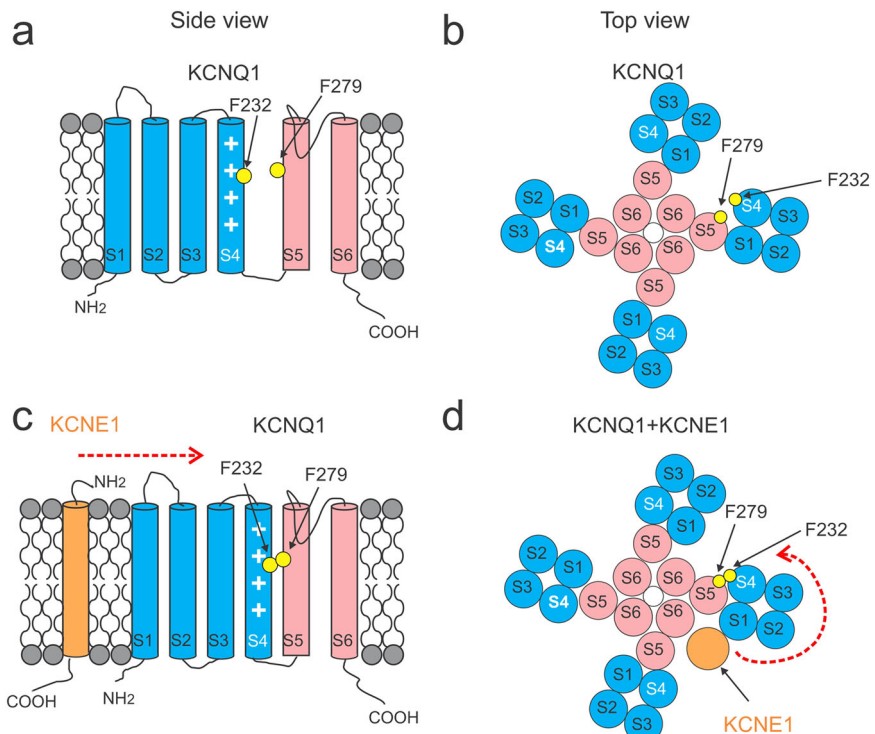

**Fig. 7 KCNE1 induces a rotation of the VSD relative to the PD of KCNQ1.** Schematic side view (**a**) and top view (**b**) of KCNQ1 channels. Schematic side view (**c**) and top view (**d**) of KCNQ1/KCNE1 channels. KCNE1 in orange imposes a rotation of the VSD compared to KCNQ1 expressed alone (indicated as red arrows). The VSD–PD interface, especially the interface between S4 and S5, would be compressed. F232 in yellow in S4 would clash with F279 in yellow in the neighboring S5 if KCNQ1 would open when S4 is in the intermediate state. This would limit the movement of PD and therefore force KCNQ1/KCNE1 channels to open only when S4 is in the fully activated state. Mutations of these residues to smaller amino acids, such as F232A and F279A, would not lead to clashes so that the intermediate-open state is allowed in the KCNQ1 channel expressed together with KCNE1.

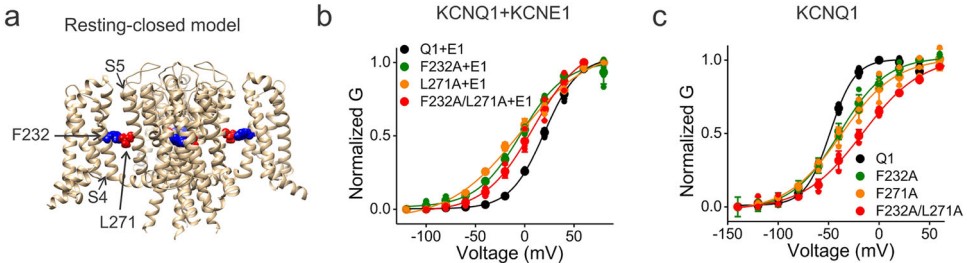

**Fig. 8 Mutant cycle analysis suggests that F232 and L271 interact in KCNQ1/KCNE1 channels but not in KCNQ1 channels alone. a** The location of F232 (blue) and L271 (red) in the resting-closed model of KCNQ1. **b** Voltage dependence of current activation from wt KCNQ1+KCNE1 (black, $n = 4$), KCNQ1-F232A/KCNE1 (green, $n = 4$), KCNQ1-L271A/KCNE1 (orange, $n = 3$), and KCNQ1-F232A-L271A/KCNE1 (red, $n = 3$) channels. **c** Voltage dependence of current activation from wt KCNQ1 (black, $n = 4$), KCNQ1-F232A (green, $n = 3$), KCNQ1-L271A (orange, $n = 3$), and KCNQ1-F232A-L271A (red, $n = 3$) channels without KCNE1 association. The voltage protocol is indicated in Fig. 2. Data are shown as mean ± SEM.

current activation of the mutant channel correlated with F2, as if the mutant channel conducts when S4 is in the fully activated state. The F275A mutant displayed a Rb$^+$/K$^+$ ratio ($0.55 \pm 0.09$) similar to wt KCNQ1/KCNE1 channels ($0.58 \pm 0.07$) (Fig. S3d), suggesting the F275A does not change the ion selectivity and does not allow KCNQ1/KCNE1 channel to open from the IO state. Interestingly, the double mutation F232A/F275A shows a significantly higher Rb$^+$/K$^+$ ratio ($1.18 \pm 0.43$), as if the F232A is dominant in the double mutant and the F232A/F275A mutant exhibits a fraction of open channels in the intermediate S4 state. The MCA (Fig. S3c, d) showed that the coupling energy for F232A and F275A in KCNQ1 and KCNQ1/KCNE1 channels is ~0.81 and ~0.66 kcal/mol, respectively, which indicates that F232 and F275 are not functionally coupled in the KCNQ1 channel in the presence or absence of KCNE1 in the intermediate S4 state,

even though they seem close in the IO model. Collectively, our VCF and Rb$^+$/K$^+$ ratio results suggest that F275A, unlike F232A, cannot change the VSD–PD coupling by opening from the IO state in KCNQ1/KCNE1 channels.

## Discussion

Using VCF and MCA, we here have shown that the F232–F279 interaction formed between the VSD from one subunit and the PD from the neighboring subunit is likely to be part of a non-canonical VSD–PD coupling in KCNQ1/KCNE1 channels. In addition, we have presented a model in which KCNE1 changes the VSD–PD coupling by imposing a rotation of the whole VSD relative to the PD of KCNQ1.

Recently, Hou and his colleagues[32] have presented a two-step VSD–PD coupling mechanism for KCNQ1 channels. The S4

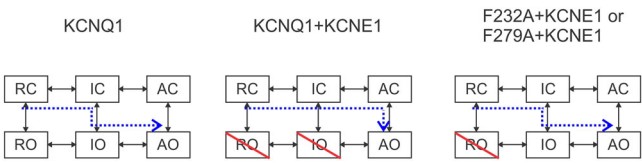

**Fig. 9 Gating model for KCNQ1, KCNQ1/KCNE1, KCNQ1-F232A/KNCE1, and KCNQ1-F279A/KNCE1 channels.** A simplified six-state model with three closed states and three open states. Horizontal transitions are two-step S4 activation while vertical transition is the channel opening. Upon activation, S4 moves from the resting state to the intermediate state and then to a fully activated state in both KCNQ1 and KCNQ1/KCNE1 channels. The KCNQ1 channel is suggested to open in both states, whereas the KCNQ1/KCNE1 channel only opens from the AO state. IO state is suppressed in KCNQ1/KCNE1 channels. F232A and F279A change the VSD–PD coupling and allow KCNQ1/KCNE1 channels to open from the IO state. RO state is suppressed in wt and mutant KCNQ1/KCNE1 channels at negative voltages. RC: resting closed; IC: intermediate closed; AC: activated closed; RO: resting-open; IO: intermediate-open; AO: activated-open.

movement from the resting state to the intermediate state leads to the IO state in which the C-terminal end of the S4–S5 linker interacts with the PD within the same subunit. The S4 movement from the intermediate state to the fully activated state leads to the AO state in which the N-terminal end of the S4–S5 linker interacts with the PD from the neighboring subunit. KCNQ1 channels expressed alone mainly open when S4 is in the intermediate state while KCNE1 was suggested to change the VSD–PD coupling of KCNQ1 so that KCNQ1/KCNE1 channels open only when S4 is in the fully activated state (Figs. 2 and 5)[19,32]. However, no molecular mechanism underlying this KCNE1-induced change in the VSD–PD coupling was proposed in these studies. We here propose that the KCNE1-induced change in the VSD–PD coupling could be partly due to the interaction formed by F232 and F279 at the interface between S4 and S5 from two different subunits. A gating model for KCNQ1, KCNQ1/KCNE1 and mutant KCNQ1/KCNE1 channels is shown in Fig. 9. Our results suggest that F232–F279 interaction is crucial for preventing the IO state of KCNQ1/KCNE1 channels and that these phenylalanines at the S4–S5 interface are important for controlling the gating of KCNQ1/KCNE1 channels. A previous study[39] has shown that F279I, a gain-of-function mutation in the KCNQ1/KCNE1 complex, causes short-QT syndrome. One possibility is that F279I allows the IO state by breaking the F232–F279 interaction and thereby creating a gain-of-function mutant of KCNQ1/KCNE1 channels. Our finding identifies the interactions contributing to the KCNE1-induced change in the VSD–PD coupling of KCNQ1/KCNE1 channels, although more interactions like F232–F279 at the VSD–PD interface need to be found to completely explain the effects of KCNE1 on KCNQ1.

The F232–F279 interaction that alters the VSD–PD coupling is located at the extracellular end of the interface between S4 and S5, which is different from the classic VSD–PD coupling that concerns the interactions between the S4–S5 linker and the intracellular end of the PD. In the canonical VSD–PD coupling model[17,18], upon depolarization, the outward movement of S4 pulls on the S4–S5 linker which in turns moves the lower end of S6 and opens the gate. However, a non-canonical VSD-PD mechanism has been recently reported in Shaker Kv channels[21–23]. These experimental and computational studies suggested that, in addition to the canonical pathway, the S4 movement triggers the channel opening via a non-canonical pathway involving the contacts between S4 and S5. Our finding is consistent with their studies that interactions between S4 and S5 are important for the VSD–PD coupling in KNCQ1/KCNE1

channels. We believe that the S4–S5 linker still plays a big role in the voltage-sensor-to-gate coupling in KCNQ1/KCNE1 channels and that the non-canonical coupling at the interface between S4 and S5 is an additional layer of control of the gating process. The interactions identified here allow the gate to open when S4 is in the intermediate state for KCNQ1 channels but not to open when S4 is in the intermediate state for KCNQ1/KCNE1 channels. Since Shaker and KCNQ1/KCNE1 channels both have a domain-swapped architecture, this non-canonical VSD–PD coupling involving VSD–PD interactions might apply to other domain-swapped voltage-gated ion channels.

We here propose a molecular mechanism of how KCNE1 suppresses the IO state of KCNQ1 channels, which until now has remained unknown. In light of the structural studies that KCNE3 imposes a counterclockwise rotation of the whole VSD relative to the PD of KCNQ1 channels[24] and functional studies that KCNE1 most likely shares a similar location with KCNE3 in the KCNQ1/KCNE complex[9], we propose that KCNE1 rotates the VSD towards to the PD of KCNQ1. This VSD rotation would force the VSD (F232 in S4) and PD (F279 in S5) closer, thereby limiting the gate movement and forcing KCNQ1 channels to open only from the AO state at more positive voltages (Figs. 7 and S2). F232A and F279A mutations can avoid the clash at the VSD–PD interface, such that the KCNQ1 channel can still open in the intermediate VSD state even with the VSD rotation induced by KCNE1 (Fig. 7). The IO state in KCNQ1/KCNE1 mutant channels seems different from that in wt KCNQ1 channels. The GV relation perfectly follows F1 in wt KCNQ1 channels (Fig. 5), but not in KCNQ1/KCNE1 mutant channels (Figs. 2 and 4). It is possible that for KCNQ1 channels, one S4 activation can open the channel when S4 moves to the intermediate state, but for KCNQ1/KCNE1 mutant channels, all four S4 activation is needed to open the channel in the S4 intermediate state.

Our proposed model that KCNE1 induces a rotation of the VSD relative to the PD and compress the VSD–PD interface is supported by our finding and other studies of KCNQ1/KCNE1 channels[26,37]. First, the F232–L271 interaction is stronger in the resting-closed state of KCNQ1 channels in the presence of KCNE1 than in the absence of KCNE1 (Fig. 8). This suggests that in the resting S4 state, F232 and L271 are physically too far apart to make strong interaction in KCNQ1 channels. Yet, KCNE1-induced rotation of the VSD would make F232 and L271 close enough to interact in KCNQ1/KCNE1 channels. Second, Nakajo and Kubo[26] found that side-chain volume at F232 and F279 when mutated to different residues showed a positive correlation with the voltage dependence of opening of KCNQ1 channels only in the presence of KCNE1. The bulkier the side-chain volume was the more positive the $V_{1/2}$, but only in the presence of KCNE1, suggesting that upon KCNE1 association the bulky residues clash at the VSD–PD interface and therefore make the gate hard to open.

Previous studies[40,41] found that V141C in KCNQ1 crosslinks with A44C in KCNE1 and that A226C in KCNQ1 crosslinks with E43C in KCNE1. However, these residues do not point towards each other if KCNE1 were positioned relative to KCNQ1 the same as KCNE3. Since KCNE1 is slightly wider than KNCE3 due to larger side chains, we hypothesize that compared to KCNE3, KCNE1 produces a larger rotation of the VSD to be able to fit into the crevice between S1 and S6. With a larger VSD rotation, residues that have been shown to form disulfide bonds would point more directly towards each other. Therefore, a KCNQ1/KCNE1 structure with this larger VSD rotation would be consistent with the data showing the formation of these disulfide bonds between KCNQ1 and KCNE1. The difference in the degree of rotation imposed by KCNE1 and KCNE3 might help understand the different regulations of KCNQ1 by different members

of the KCNE family. We have previously shown that KCNE3 only affects S4 movement through an electrostatic effect of the two negatively charged residues at the top of the TM of KCNE3 with the top S4 positive charges[42]. KCNE3 only indirectly affects the gate opening through the VSD–PD coupling. The smaller rotation of the VSD by KCNE3 would not allow for much of an effect of KCNE3 on the interface between S4 and S5 and would therefore not change the VSD–PD coupling and prevent opening from the intermediate state[30]. On the other hand, KCNE1 does not have the two negative charges on top of the TM as KCNE3 and therefore does not have the large effect on S4 movement as KCNE3. However, a larger rotation of the VSD by the larger KCNE1 would affect the interface between S4 and S5, change the VSD–PD coupling, and thereby inhibit opening from the intermediate state.

More recently, a structural study of human KCNQ2 (a paralog of KCNQ1) channels[43] revealed that a small-molecule activator (ztz-240) binds to the KCNQ2 channel and induces a similar rotation of the VSD in KCNQ2 channels that we propose here for KCNQ1/KCNE1 channels. The rotation of the VSD was suggested to enhance interactions at the interface between S4 and S5, such as the F202–F248 interaction in KCNQ2 channels (homologous to F232–F279 interaction in KCNQ1). We, therefore, believe this mechanism is likely to apply to other Kv channels for which modulatory subunits and/or drugs might rotate the VSD and alter VSD–PD interactions, thereby affecting the gating of Kv channels.

Different groups have proposed mechanisms of how KCNE1 modulates the function of KCNQ1[19,27,34,38,40,44]. The Cui lab[19] concluded that most, if not all, of the effects of KCNE1 are due to changes of the VSD–PD coupling, although no molecular mechanism of how KCNE1 changes the VSD–PD coupling was suggested. Our proposed model of a KCNE1-induced rotation of the VSD that alters the VSD–PD coupling could potentially explain most of the effects of KCNE1 on KCNQ1 gating and permeation. We here found that F232A and F279A at the VSD–PD interface reduce many effects of KCNE1 on KCNQ1. For example, F232A and F279A remove the KCNE1 inhibition of opening of the channel when S4 is in the intermediate state, abolish the KCNE1-induced change in $Rb^+/K^+$ selectivity ratio, and reduces the KCNE1-induced positive shift in the GV relation. Kubo's group has previously shown that these mutations reduce the KCNE1-induced slowing of the activation kinetics and the voltage shifts of the GV relation[26]. The normal inhibition of the IO state by the KCNE1-induced rotation of the VSD would explain the KCNE1-induced +50 mV GV shift, the slow of current activation kinetics, and the change in $Rb^+/K^+$ selectivity by only allowing KCNQ1/KCNE1 channels to open to the AO state at more positive voltages. In addition, KCNE1 was shown to inhibit the resting-open (RO) state of KCNQ1 channels at negative voltages[16,45]. The stronger interaction between F232 and L271 in the resting state could explain the KCNE1-induced reduction in the open probability of KCNQ1 channels at negative voltages by inhibiting the movement of S5 and S6 to open the gate (Figs. 9 and S2j).

## Methods

**Molecular biology**. All mutations were introduced using QuikChange site-directed mutagenesis kit (Qiagen). In vitro KCNQ1 and KCNE1 cRNA were transcribed using mMessage mMachine T7 RNA Transcription Kit (Ambion).

**Two-electrode voltage clamp**. A total of 50 ng of *KCNQ1* cRNA was injected into defolliculated *Xenopus laevis* oocytes (Ecocyte, Austin, TX). For KCNQ1/KCNE1 experiments, oocytes were co-injected with *KCNQ1* cRNA and *KCNE1* cRNA by a ratio of 3:1, weight:weight. After cRNA injection, oocytes were incubated in ND96 solution (96 mM NaCl, 2 mM KCl, 1 mM MgCl₂, 1.8 mM CaCl₂, 5 mM HEPES; pH = 7.5) for 2–5 days before electrophysical experiments. $Rb^+/K^+$

permeability experiments were recorded in 100 mM Na⁺ solution (96 mM NaCl, 4 mM KCl, 1.8 mM CaCl₂, 1 mM MgCl₂, 5 mM HEPES; pH = 7.5), 100 mM K⁺ solution (100 mM KCl, 1.8 mM CaCl₂, 1 mM MgCl₂, 5 mM HEPES; pH = 7.5) or 100 mM Rb⁺ solution (96 mM RbCl, 4 mM KCl, 1.8 mM CaCl₂, 1 mM MgCl₂, 5 mM HEPES; pH = 7.5); 5 μM XE991 (Tocris Bioscience) was added to the bath solution.

**Voltage clamp fluorometry (VCF)**. After 2–5-day incubation, oocytes were labeled with 100 μM Alexa-488 maleimide (Molecular Probes) for 30 min at 4 °C. Following labeling, they were kept on ice to prevent internalization of labeled channels. Oocytes were recorded in ND96 solution. Then, 100 μM LaCl₃ were used to block endogenous currents induced by hyperpolarized voltages and has no effects on GV and FV curves from KCNQ1 and KCNQ1/KCNE1 channels[27]. From a holding potential of −120 mV, steps from −160 mV (only traces from −140 mV are shown in the current recordings due to endogenous currents at very negative voltages) to +80 mV (in +20 mV steps) were applied to activate the S4 movement and current of the channel followed by a tail voltage of −40 mV to obtain the tail current.

**Thermodynamic mutant cycle analysis**. The amount of free energy required to shift the channel from the open to closed state was calculated as $\Delta G^0_{O \to C} = -zFV_{1/2}$, where $z$ and $V_{1/2}$ were obtained from the $G(V)$, $F$ is the Faraday's number. The perturbation in free energy by a mutation relative to the wt was calculated as $\Delta\Delta G^0 = \Delta(zFV_{1/2}) = -F(z^{wt}V^{wt}_{1/2} - z^m V^m_{1/2})$. The coupling energy between each pair of residues was calculated as $\Delta G^0_{coupling} = \Delta\Delta G^0_1 + \Delta\Delta G^0_2 - \Delta\Delta G^0_{1,2} = -F(z^{wt}V^{wt}_{1/2} - z^{m1}V^{m1}_{1/2}) - F(z^{wt}V^{wt}_{1/2} - z^{m2}V^{m2}_{1/2}) + F(z^{wt}V^{wt}_{1/2} - z^{m1,2}V^{m1,2}_{1/2}) = -F(z^{wt}V^{wt}_{1/2} - z^{m1}V^{m1}_{1/2} - z^{m2}V^{m2}_{1/2} + z^{m1,2}V^{m1,2}_{1/2})$.

**Molecular modeling**. Models were created by recent structures[24,30] and models[25] of KCNQ1 and KCNE beta-subunits as templates. A homology model of KCNE1 was created from KCNE3 in the KCNQ1/KCNE3 cryo-EM structure using the Swiss-model program (https://swissmodel.expasy.org/). Homology models of IC and IO states of the human KCNQ1/KCNE1 were created by replacing the VSD in the closed and open KCNQ1/KCNE3 cryo-EM structure, respectively, with the VSD in the KCNQ1 NMR structure. KCNE1 was initially placed in the same position as KCNE3 in the KCNQ1/KCNE3 cryo-EM structure. However, clashes between side chains of KCNE1 and KCNQ1 required additional changes. We rotated the VSD of KCNQ1 an additional 20° counterclockwise (seen from the extracellular side) compared to the KCNQ1/KCNE3 structure and rotated and translated KCNE1 to better fit in S1 and S5 in KCNQ1 and to better agree with previous disulfide bridges between KCNE1 and S1[41,46]. Models of IC and IO states of human KCNQ1 were created by replacing the VSD in the closed and open KCNQ1 cryo-EM structure, respectively, with the VSD in the KCNQ1 NMR structure. For the PDB of KCNQ1/KCNE1 IO model, CHARMM-GUI membrane builder protocol (steps 5–7)[47] was used to prepare full assembly embedded into a POPC (16:0–18:1 PC 1-palmitoyl-2-oleoyl-sn-glycero-3-phosphocholine) lipid bilayer and solvated in 150 mM KCl aqueous solution using the latest CHARMM-36 force field for proteins and lipids[48,49], default NBFIX values for ions[50] and TIP3P water model[51]. The final simulation system contains ~220,000 atoms including solvent molecules and counterions. The standard C-term and N-term capping was used for each of the protomers. The protonation state was set up to residue standard protonation at pH = 7.4, accordingly. The fully assembled system was equilibrated for 50 ns with a harmonic constraints applied to backbone heavy atoms using NAMD2.15 GPU-based software[52]. During this equilibration stage, constraints were gradually reduced from 10 kcal/mol/Å² to 0 kcal/mol/Å² for the subsequent production run. The production run was performed for 250 ns to further refine developed model.

**Statistics and reproducibility**. GV curve was obtained by plotting the normalized tail currents versus different test pulses to determine the steady-state voltage dependence of current activation. Tail currents were measured at −40 mV following test pulses. Then GV curve was fit with a single Boltzmann equation: $G(V) = A_{min} + (A_{max} - A_{min})/(1 + \exp((V - V_{1/2})/K))$, where $A_{max}$ and $A_{min}$ are the maximum and minimum, respectively, $V_{1/2}$ is the voltage where 50% of the maximal conductance level is reached and $K$ is the slope factor. Data were normalized between the $A_{max}$ and $A_{min}$ values of the fit. Fluorescence signals were bleach-subtracted, and data points were averaged over tens of milliseconds at the end of the test pulse to reduce errors from signal noise. FV curve was obtained by plotting the normalized steady-state fluorescence signal versus different test pulses. Then FV curve was fit with a double Boltzmann equation.

All experiments were repeated more than three times from at least two batches of oocytes. Statistical data analysis was performed using Student's t-test or ANOVA with a Tukey's test. Data are presented as mean ± SEM, and N represents the number of experiments.

**Reporting summary**. Further information on research design is available in the Nature Research Reporting Summary linked to this article.

## Data availability

All data generated or analyzed during this study are included in this manuscript. The source data is provided with this paper as Supplementary Data 2. All other data are available from the corresponding author on reasonable request.

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

## Acknowledgements

The research in the Larsson lab is funded by R01 HL131461 and R01 GM109762. We thank Barro-Soria, University of Miami, for helpful comments on the manuscript.

## Author contributions

X.W. and H.P.L. designed the research. X.W. and M.E.P. performed experiments. X.W., S.Yu.N., and H.P.L analyzed the data and wrote the manuscript.

## Competing interests

The authors declare no competing interests.
