## [Peer Review File. · Communications Biology]

Reviewers' Comments:

Reviewer #1:

Remarks to the Author:

This is a very interesting manuscript that deals with the molecular coupling of the voltage-sensing and pore domains of KCNQ1 in the presence of KCNE1. Manuscript is clearly written and data is mostly compelling.

Major comment:

1) The data presented suggest that the mutations allow opening of the KCNQ1/KCNE1 channel in the intermediate stage. Nonetheless, mutations affected the fluorescence-voltage relationship. The authors indicate (line 90) in the text that the voltage-dependence of the FV relationship is not affected, while later (line 107), suggest the F1 is shifted, consistent the figure presented. Please clarify the statements. Please show further analysis of the FV relationship (both F1 and F2 in detail) and comment on the consequences of any effects observed in the FV relationship caused by the F232A mutation. Could the shift on F1 caused by the mutation be the result of the shift in the GV observed?

2) The authors suggest that F232A and F279A mutations show similar GV and FV relations (line 175). That does not seem to be the case when comparing figure 2B and figure 4B. Please comment. The lack of a good fit for the F1/F2 components in figure 4B makes it difficult to compare the GV curve the F1 component.

3) It would be useful to see the effect of F232A and F279A mutations on KCNQ1 Rb permeability.

4) Please show a more detailed comparison between the effect of the mutations on KCNQ1 FV and GV relations. Please comment on how the opening in the intermediate stage for the KCNQ1/KCNE1 mutant channel compares with the opening of the WT KCNQ1 channel in the intermediate state. It seems that the channel is not open to the same extent as the WT KCNQ1 channel, which is not a problem, but should be discussed.

5) It would be preferable if in Figure 6C the errors were smaller. It would increase confidence on the effect observed.

Reviewer #2:

Remarks to the Author:

This interesting study explores the mechanism by which KCNE1 modulates the properties of KCNQ1, a potassium channel that plays a critical role in the repolarization phase of the ventricular action potential. Convincing evidence is presented that points to a role of KCNE1 in regulating a new type of coupling between voltage-sensing domain (VSD) and pore domain (PD) via rotation of the VSD relative to the PD. This KCNE1-induced rotation is proposed to destabilize the interaction between at least two phenylalanines at the VSD-PD interface, thus preventing the channel from opening when the VSD is not in the fully activated conformation. The findings are novel and will be of interest to a wide community. Overall, the functional measurements were carefully performed and their interpretation is clear. The structural modeling, on the other hand, needs clarifications on several points.

Mutation F232A in the channel S4 helix is found to allow the KCNQ1-KCNE1 complex to open before the VSD can reach the fully activated state. This is first deduced by the fact that the second component of the FV relation (F2) is right-shifted compared to the GV relation (Fig. 2B). This shift is not particularly large and its assessment is mostly based on a couple of points. The GV and FVs of the mutant complex should be determined at higher resolution (e.g., steps every 10 mV) in the range -40 mV to +80 mV to better substantiate this important claim.

The subsection on page 13 is titled "KCNE1 induces a rotation of the VSD relative to the PD and

inhibits the intermediate open state”, which leads to the expectation that this rotation is demonstrated in this section. However, the section presents a hypothesis based on recent structural work by the MacKinnon lab on the KCNQ1-KCNE3 complex. The title should mention the hypothesis and the structural models that were generated to support it.

It is not clear how the rotation of the VSD was built into the models, especially in the case where the KCNQ1 VSD from the Cryo-EM structure was substituted with the VSD from the NMR structure. More information on how the models were built is required.

In the discussion, it is stated that “compared to KCNE3, KCNE1 produces a larger rotation...” how was this larger rotation accounted for in the models if they were based on the complex with KCNE3?

The legend for Fig. 9 states that this is a “five-state [gating] model” when, in fact, there are six states. Did the authors mean to cross the RO state also from KCNQ1 gating? This mechanism raises another question about the KCNQ1-KCNE3 complex used to build the structural models with KCNE1. The introduction states that “KCNE3 removes the voltage dependence of KCNQ1 current activation, locking the channel open in the physiological voltage range”. If both KCNE1 and KCNE3 produce similar rotations of the VSD relative to the PD, why do they have such different effects on VSD-PD coupling?

Minor points:

The first paragraph of the Discussion states that “...we have presented a novel mechanism by which KCNE1 changes the VSD-PD coupling by imposing a rotation of the whole VSD relative to the PD of KCNQ1.” This should say “...we have proposed a new model” as the rotation of the VSD is inferred, not demonstrated.

Line 413: “rotation of the VSD to be able to fit into the crevice”

Lines 414-415 need rewording: “residues in both of these disulfide bonds would point towards each other and would be more likely to form disulfide bonds between KCNQ1 and KCNE1.”

Reviewer #3:

Remarks to the Author:

Exactly how the KCNE1 protein modulates the KCNQ1 potassium channel, as is required for formation of the critical I_{Ks} current of the cardiac action potential, is an important longstanding and unresolved problem. A series of models for how KCNE1 shifts V_{1/2} for channel activation, slows down channel activation, enhances maximum conductance, and eliminates channel inactivation have been generated over the years, starting with C-Kang et al. in 2008. While these models certainly have advanced the field, none of them have held up well in the face of additional data. This manuscript from Wu, Perez, and Larsson presents data that leads to what I find to be a truly compelling model for how KCNE1 works. Specifically, mutagenesis data is combined with voltage-clamp fluorimetry and electrophysiology to demonstrate that KCNE1 causes rotation of the voltage sensor domain KCNQ1 relative to the pore domain, with the consequence that residue clashes occur that prohibit formation of a well-established conductive intermediate state conformation of the voltage sensor. This is why the channel is conductive only in the “activated open” state of the channel when KCNE1 is present. This model was underpinned in this work by finding mutations that allow formation of the conductive intermediate state even when KCNE1 is bound. This work is very well written and the story is compelling. I think this work should certainly be published, but do encourage that the authors address the following suggestions through minor revision:

(1) The PDB coordinates for their intermediate state KCNE1/KCNQ1 model should be included as a supporting file to this document and made available at the journal web site.

(2) While I think their model for KCNE1 action is very compelling, exactly how interactions of the key residues F232 and F239 are coupled to channel gating is, as far as I can tell, not clear. For

example, could it not be that gating still critically involves the S4 control of S4-S5 interactions with the gate, but that interactions of the upper parts of the voltage sensor domain and the pore domain are coupled to what is happening at the S4-S5 linker? A good summary stating what is clearly established and what remains mysterious in terms of a comprehensive model of KCNE1 action would be helpful and will help guide future work.

(3) There is a 2020 eLife paper by G-Kuenze et al that presents a lot of new data and a new model for KCNE1/KCNQ1 interactions that should be cited and perhaps discussed. Unfortunately, that work overlooked the intermediate state as possibly being key to understanding the role of KCNE1 and so the modeling in that work may be useful, but does not get to the root of the matter.

(4) It is known that the F279I mutation in KCNQ1 is a gain-of-function mutation that causes Short QT Syndrome (Morena et al., 2015). I think this previous observation further supports the results of this paper, as seems worth pointing out and citing.

Again, I emphasize that this is beautiful work that should be published expeditiously.

We thank the reviewers for their constructive criticism and suggestions. We have responded to all of their comments. We think this has further strengthened the manuscript.

Reviewer #1 (Remarks to the Author):

This is a very interesting manuscript that deals with the molecular coupling of the voltage-sensing and pore domains of KCNQ1 in the presence of KCNE1. Manuscript is clearly written and data is mostly compelling.

Major comment:

1) The data presented suggest that the mutations allow opening of the KCNQ1/KCNE1 channel in the intermediate stage. Nonetheless, mutations affected the fluorescence-voltage relationship. The authors indicate (line 90) in the text that the voltage-dependence of the FV relationship is not affected, while later (line 107), suggest the F1 is shifted, consistent the figure presented. Please clarify the statements. Please show further analysis of the FV relationship (both F1 and F2 in detail) and comment on the consequences of any effects observed in the FV relationship caused by the F232A mutation. Could the shift on F1 caused by the mutation be the result of the shift in the GV observed?

- We have corrected the statements about the shift in F1. We have also added more analysis of the FVs and GVs in the table below and added some comments about the effects of F232A on the FV. Compared to wt, F232A shifted the GV to the negative voltages and slightly shifted the F1. The fact that KCNQ1-F232A/KCNE1 channels can open after the first S4 movement could be the reason for the slight shift in F1, because for F232A there is an additional state (compared to wt) with S4 in the intermediate state (IO). This most likely should further stabilize the intermediate state of S4 and give rise to the small shift in F1.

	$G_{1/2}$ (mV)	$F1_{1/2}$ (mV)	$F2_{1/2}$ (mV)
Wt KCNQ1/KCNE1	21.58 ± 0.56	-86.51 ± 4.49	50.14 ± 3.52
KCNQ1-F232A/KCNE1	-2.57 ± 1.61	-97.40 ± 1.65	49.94 ± 4.82

Table 1. $G_{1/2}$ is the voltage where 50% of the maximal conductance is reached. Likewise, $F1_{1/2}$ is the voltage where 50% of the first component of fluorescence is reached and $F2_{1/2}$ is the voltage where 50% of the second component of fluorescence is reached.

2) The authors suggest that F232A and F279A mutations show similar GV and FV relations

(line 175). That does not seem to be the case when comparing figure 2B and figure 4B. Please comment. The lack of a good fit for the F1/F2 components in figure 4B makes it difficult to compare the GV curve the F1 component.

- We have corrected this statement and made better recordings and fits in Fig 4B.

3) It would be useful to see the effect of F232A and F279A mutations on KCNQ1 Rb permeability.

- We have added data on F232A and F279A Rb/K permeability ratio in KCNQ1 channels without KCNE1, which is the Supplementary figure 3. We found that these two mutations did not significantly change the Rb/K ratio.

4) Please show a more detailed comparison between the effect of the mutations on KCNQ1 FV and GV relations. Please comment on how the opening in the intermediate stage for the KCNQ1/KCNE1 mutant channel compares with the opening of the WT KCNQ1 channel in the intermediate state. It seems that the channel is not open to the same extent as the WT KCNQ1 channel, which is not a problem, but should be discussed.

- We have added more analysis of the FVs and GVs of KCNQ1 mutations in the table below. In wt KCNQ1 channels alone, the GV relation overlaps with F1 (Fig. 5) and the $G_{1/2}$ is very close to $F1_{1/2}$, suggesting KCNQ1 opens when S4 is in the intermediate state. Compared to wt KCNQ1 channels, F232A (without KCNE1) shifted the GV and F1 in the same direction, with GV tightly follows F1 ($G_{1/2}$ and $F1_{1/2}$ are similar). That means F232A opens in the

intermediated S4 state. F279A shifted the GV, F1 and F2 to more negative voltages. Also, $G_{1/2}$ and $F1_{1/2}$ for F279A is very similar, suggesting F279A opens from the intermediate-open state.

	$G_{1/2}$ (mV)	$F1_{1/2}$ (mV)	$F2_{1/2}$ (mV)
Wt KCNQ1	-49.30 ± 1.39	-47.88 ± 0.68	58.30 ± 15.57
F232A	-39.40 ± 1.61	-40.07 ± 1.68	48.95 ± 3.24
F279A	-57.13 ± 2.96	-56.51 ± 2.23	36.37 ± 5.78

For the comparison of the intermediate state between KCNQ1-F232A/KCNE1 and wt KCNQ1 channels, there are clear differences. The GV relation perfectly follows F1 in wt KCNQ1 channels, but not in KCNQ1-F232A/KCNE1 channels. It is possible that for KCNQ1 channels, one S4 activation can open the channel when S4 moves to the intermediate state, but for KCNQ1-F232A/KCNE1 channels, all four S4 activation is needed for the intermediate-open state. This could be one possible reason for the difference in $F1_{1/2}$ and $G_{1/2}$ for KCNQ1 F232A/KCNE1 channels. We have included this in the discussion.

	$G_{1/2}$ (mV)	$F1_{1/2}$ (mV)	$F2_{1/2}$ (mV)
Wt KCNQ1	-49.30 ± 1.39	-47.88 ± 0.68	58.30 ± 15.57
KCNQ1-F232A/KCNE1	-2.57 ± 1.61	-97.40 ± 1.65	49.94 ± 4.82

5) It would be preferable if in Figure 6C the errors were smaller. It would increase confidence on the effect observed.

- We have increased the number of cells to reduce the error bars.

Reviewer #2 (Remarks to the Author):

This interesting study explores the mechanism by which KCNE1 modulates the properties of KCNQ1, a potassium channel that plays a critical role in the repolarization phase of the ventricular action potential. Convincing evidence is presented that points to a role of KCNE1 in regulating a new type of coupling between voltage-sensing domain (VSD) and pore domain (PD) via rotation of the VSD relative to the PD. This KCNE1-induced rotation is proposed to destabilize the interaction between at least two phenylalanines at the VSD-PD interface, thus preventing the channel from opening when the VSD is not in the fully activated conformation. The findings are novel and will be of interest to a wide community. Overall, the functional measurements were carefully performed and their interpretation is clear. The structural

modeling, on the other hand, needs clarifications on several points.

1) Mutation F232A in the channel S4 helix is found to allow the KCNQ1-KCNE1 complex to open before the VSD can reach the fully activated state. This is first deduced by the fact that the second component of the FV relation (F2) is right-shifted compared to the GV relation (Fig. 2B). This shift is not particularly large and its assessment is mostly based on a couple of points. The GV and FVs of the mutant complex should be determined at higher resolution (e.g., steps every 10 mV) in the range -40 mV to +80 mV to better substantiate this important claim.

- We have now recorded the FV and GV at higher resolution with 10 mV intervals. F2 is right-shifted compared to the GV relation, suggesting that F232A allows the KCNQ1-KCNE1 channel to open before the VSD can reach the fully activated state.

2) The subsection on page 13 is titled "KCNE1 induces a rotation of the VSD relative to the PD and inhibits the intermediate open state", which leads to the expectation that this rotation is demonstrated in this section. However, the section presents a hypothesis based on recent structural work by the MacKinnon lab on the KCNQ1-KCNE3 complex. The title should mention the hypothesis and the structural models that were generated to support it.

- The title has been changed accordingly.

3) It is not clear how the rotation of the VSD was built into the models, especially in the case where the KCNQ1 VSD from the Cryo-EM structure was substituted with the VSD from the NMR structure. More information on how the models were built is required.

- More details of how the KCNQ1/KCNE1 model was constructed have been incorporated in the methods.

4) In the discussion, it is stated that "compared to KCNE3, KCNE1 produces a larger rotation..." how was this larger rotation accounted for in the models if they were based on the complex with KCNE3?

- The larger rotation was based on the side chain clashes when KCNE1 was introduced in place of KCNE3 due to the larger side chains of KCNE1 compared to KCNE3. In addition, the larger rotation is also more consistent with previously reported disulfide bonds between KCNQ1 and KCNE1. (DOI: 10.1085/jgp.201110672)

We imagine that the different sizes of VSD rotation in KCNE3 vs KCNE1 plays a role in the different effects. We have previously shown that KCNE3 only affects S4 movement through an electrostatic effect of the two negatively-charged residues at the top of the TM of KCNE3 with the top S4 positive charges. KCNE3 only indirectly affects gate opening through the voltage-sensor-to-gate coupling. The smaller rotation of the VSD by KCNE3 would not allow for much of an effect of KCNE3 on the S4-S5 interface and would therefore not change the voltage-sensor-to-gate coupling and prevent opening from the intermediate state. On the other hand, KCNE1 does not have the two negative charges on top of the TM as KCNE3 and therefore does not have the large effect on S4 movement as KCNE3. However, a larger rotation of the VSD by the larger KCNE1 would affect the S4-S5 interface, change the voltage sensor-to-gate coupling, and thereby inhibit opening from the intermediate state.

5) The legend for Fig. 9 states that this is a "five-state [gating] model" when, in fact, there are six states. Did the authors mean to cross the RO state also from KCNQ1 gating? This mechanism raises another question about the KCNQ1-KCNE3 complex used to build the structural models with KCNE1. The introduction states that "KCNE3 removes the voltage dependence of KCNQ1 current activation, locking the channel open in the physiological voltage range". If both KCNE1 and KCNE3 produce similar rotations of the VSD relative to the PD, why do they have such different effects on VSD-PD coupling?

- The model is now referred to as 6-state model.

Minor points:

1) The first paragraph of the Discussion states that "...we have presented a novel mechanism by which KCNE1 changes the VSD-PD coupling by imposing a rotation of the whole VSD

relative to the PD of KCNQ1." This should say "...we have proposed a new model" as the rotation of the VSD is inferred, not demonstrated.

- The text has been changed accordingly.

2) Line 413: "rotation of the VSD to be able to fit into the crevice"

- The text has been changed accordingly.

3) Lines 414-415 need rewording: "residues in both of these disulfide bonds would point towards each other and would be more likely to form disulfide bonds between KCNQ1 and KCNE1."

- The text has been changed to "With a larger rotation of the VSD, residues that that have been shown to form disulfide bonds would point more directly towards each other. Therefore, a KCNQ1/KCNE1 structure with this larger VSD rotation would be consistent with the data showing the formation of these disulfide bonds between KCNQ1 and KCNE1."

Reviewer #3 (Remarks to the Author):

Exactly how the KCNE1 protein modulates the KCNQ1 potassium channel, as is required for formation of the critical I_{Ks} current of the cardiac action potential, is an important longstanding and unresolved problem. A series of models for how KCNE1 shifts V_{1/2} for channel activation, slows down down channel activation, enhances maximum conductance, and eliminates channel inactivation have been generated over the years, starting with C-Kang et al. in 2008. While these models certainly have advanced the field, none of them have held up well in the face of additional data. This manuscript from Wu, Perez, and Larsson presents data that leads to what I find to be a truly compelling model for how KCNE1 works. Specifically, mutagenesis data is combined with voltage-clamp fluorimetry and electrophysiology to demonstrate that KCNE1 causes rotation of the voltage sensor domain KCNQ1 relative to the pore domain, with the consequence that residue clashes occur that prohibit formation of a well-established conductive intermediate state conformation of the voltage sensor. This is why the channel is conductive only in the "activated open" state of the channel when KCNE1 is present. This model was underpinned in this work by finding mutations that allow formation

of the conductive intermediate state even when KCNE1 is bound. This work is very well written and the story is compelling. I think this work should certainly be published, but do encourage that the authors address the following suggestions through minor revision:

(1) The PDB coordinates for their intermediate state KCNE1/KCNQ1 model should be included as a supporting file to this document and made available at the journal web site.

- The PDB coordinates are now available.

(2) While I think their model for KCNE1 action is very compelling, exactly how interactions of the key residues F232 and F239 are coupled to channel gating is, as far as I can tell, not clear. For example, could it not be that gating still critically involves the S4 control of S4-S5 interactions with the gate, but that interactions of the upper parts of the voltage sensor domain and the pore domain are coupled to what is happening at the S4-S5 linker? A good summary stating what is clearly established and what remains mysterious in terms of a comprehensive model of KCNE1 action would be helpful and will help guide future work.

- We agree that the S4-S5 linker still plays a big role and that the S4-S5 interface is an additional layer of control of the gating process. Previous work by the Cui group has identified residue interactions between the S4-S5 linker (or lower S4) and the pore (lower S5 and S6) that are important for the voltage sensor-to-gate coupling (DOI: 10.1038/s41467-020-14406-w). We believe that the interactions identified here are an additional control of the gate, which allows the gate to open when S4 is in the intermediate state for KCNQ1 channels but not to open when S4 is in the intermediate state for KCNE1/KCNQ1 channels. This is now better explained in the Discussion.

(3) There is a 2020 eLife paper by G-Kuenze et al that presents a lot of new data and a new model for KCNE1/KCNQ1 interactions that should be cited and perhaps discussed. Unfortunately, that work overlooked the intermediate state as possibly being key to understanding the role of KCNE1 and so the modeling in that work may be useful, but does not get to the root of the matter.

- We now refer to the Kuenze 2020 paper in the Discussion.

(4) It is known that the F279I mutation in KCNQ1 is a gain-of-function mutation that causes Short QT Syndrome (Morena et al., 2015). I think this previous observation further supports the

results of this paper, as seems worth pointing out and citing.

- Thank you for the reference. We now include this in the Discussion.

Again, I emphasize that this is beautiful work that should be published expeditiously.

Reviewers' Comments:

Reviewer #1:

Remarks to the Author:

The authors responded to all my concerns satisfactorily.

Reviewer #2:

Remarks to the Author:

All my concerns were properly addressed.

Reviewer #3:

Remarks to the Author:

I think the authors have done an excellent job addressing the reviewer concerns and recommend acceptance of this paper.